# Verification of Footwear Effects on a Foot Deformation Approach for Estimating Ground Reaction Forces and Moments

**DOI:** 10.3390/s25123705

**Published:** 2025-06-13

**Authors:** Naoto Haraguchi, Hajime Ohtsu, Bian Yoshimura, Kazunori Hase

**Affiliations:** 1Department of Mechanical Systems Engineering, Tokyo Metropolitan University, Hino 191-0065, Japan; ohtsu.hajime.es@osaka-u.ac.jp (H.O.); yoshimura.bian@gmail.com (B.Y.); 2Department of Assistive Technology, Research Institute of National Rehabilitation Center for Persons with Disabilities, Tokorozawa 359-8555, Japan; 3Department of Mechanical Science and Bioengineering, The University of Osaka, Toyonaka 560-8531, Japan

**Keywords:** gait analysis, optical motion capture, inverse kinematics, dynamics analysis

## Abstract

The foot deformation approach (FDA) estimates the ground reaction force (GRF) and moment (GRM) from kinematic data with practical accuracy, low computational cost, and no requirement for training data. Our previous study demonstrated practical estimation accuracy of the FDA under barefoot conditions. However, since the FDA estimates GRFs and GRMs based on foot deformation under body weight, there are concerns about its applicability to footwear conditions, where the foot deformation characteristics differ from those of bare feet. Following the issue, this study conducted a walking experiment at three different speeds with running shoes and sneakers to investigate the impact of footwear on GRF prediction using the FDA. The results showed that the FDA successfully provided practical accuracy when shoes were worn, comparable to that for a barefoot participant. The FDA offers advantages for estimating GRFs and GRMs for the footwear condition, while eliminating the need for collecting training data and enabling rapid analysis and feedback in clinical settings. Although the FDA cannot fully eliminate the effects of footwear and movement speed on prediction accuracy, it has the potential to serve as a convenient biomechanical-based method for estimating GRFs and GRMs during sports and daily activities with footwear.

## 1. Introduction

The ground reaction force (GRF) and moment (GRM) are crucial factors in biomechanical studies, such as assessing the risk of musculoskeletal disorders and evaluating physical performance [1,2]. GRFs and GRMs are typically measured using force plates, which directly record the forces acting on the human foot. However, because force plates must be embedded in the ground, they have limitations regarding measurement locations, which restricts their applicability in clinical settings. While wearable recording systems, such as pressure-sensing insoles and force-sensing shoes [3], have been developed to enable measurements in more versatile environments, they still face challenges associated with the wearable sensors, including low durability and the negative effect of device weight on natural human movement. To address these issues, researchers have developed prediction techniques for the GRF and GRM using kinematic data recorded by optical motion capture (OMC) or inertial measurement units [4]. Prediction techniques that monitor GRFs and GRMs without force sensors contribute to simplifying biomechanical analysis in clinical settings.

The mainstream approach for GRF prediction involves machine learning, which offers the advantage of estimating GRFs and GRMs with high accuracy and fast computational time by leveraging the statistical relationship between GRFs/GRMs and human kinematics [5,6,7]. However, machine learning methods—such as GRF prediction using artificial neural networks—require a large amount of training data to achieve high reliability in prediction [4]. Consequently, while machine learning-based approaches are effective for GRF prediction in healthy participants or during typical human movements, they present challenges due to the high cost of data collection in situations where collecting sufficient training data is difficult, such as with elderly and disabled persons with physical disfunctions, or during specialized sports activities. Therefore, machine learning-based methods do not fully address all scenarios for GRF prediction. As an alternative approach that does not rely on training data, biomechanical model-based methods have been employed in the development of GRF prediction techniques [4].

A biomechanical model-based technique has been developed as another approach for estimating GRFs and GRMs, relying on biomechanical assumptions regarding human activities [4]. This method employs computational techniques based on biomechanics, such as musculoskeletal and contact models, without the need for training data, thereby addressing gaps not covered by the machine learning-based approach. However, the biomechanical model-based approach faces the challenge of high computational costs due to the optimization processes required to tune parameters for musculoskeletal and contact models [8,9,10,11,12,13]. This limitation poses a disadvantage for performing efficient analysis and providing rapid feedback on the estimation results to researchers and participants, thereby preventing the broader application of the biomechanical model-based estimation to clinical settings.

To address the issue of the biomechanical model-based estimation, we have developed a new biomechanical model-based approach for estimating GRFs and GRMs by focusing on the deformation in foot alignment obtained by OMC, named the foot deformation approach (FDA) [14]. This method estimates the distribution of forces acting on the foot based on its deformations under the load of the body mass. Our FDA successfully estimated the GRF and GRM with practical accuracy and no requirement for training data and an optimization process. This approach overcomes the challenges of data collection cost in the machine learning-based estimation and computational cost in the current biomechanical model-based estimation. However, the impact of footwear on prediction accuracy in the FDA needs further investigation, as the elasticity of shoe uppers and soles may affect the characteristics of foot deformation. In many sports and daily activities, biomechanical analyses involving GRFs have been performed under footwear conditions [15,16]. Therefore, this study initially analyzed the effect of footwear on the estimation accuracy of the FDA, aiming to verify its applicability for GRF prediction when participants use footwear during normal activities. Walking experiments were performed, whose practical estimation accuracy under barefoot conditions had already been demonstrated in a previous study [14], and were used to investigate whether the FDA provides estimation accuracy comparable to that of the previous studies by comparing the FDA estimation under footwear conditions with that of the current biomechanical model-based method.

## 2. Materials and Methods

### 2.1. Estimation Algorithm

In this study, GRFs and GRMs have been estimated based on the FDA, as described in our previous study [14]. Figure 1 shows the computational flow of the FDA. First, the OMC system measures human movements and computes the acceleration of body segments, the position of anatomical landmarks, and the position of the center of mass (COM) based on the Gait 2392 musculoskeletal model in OpenSim [17,18,19,20].

After the kinematics data are collected, the total external force acting on the human body is calculated using the translational equations of motion. The inertia parameters required for the dynamics equations are defined by the Gait 2392 musculoskeletal model in OpenSim [17,18,19,20]. In this computational approach, the rotational equation of motion was excluded to eliminate modeling errors in inertial parameters, such as the moment of inertia and the center of mass of each segment. Instead, GRFs and GRMs were estimated based on external forces derived from the translational equation of motion and their allocation using contact point analysis.

Foot contact points are defined based on anatomical landmarks on the foot segments with affixed OMC markers, for computing the distance between the contact points and the ground surface. In this study, the contact point locations were determined based on a previous study [14], with some modifications to improve estimation accuracy. To account for the medial load exerted by the shoe sole supporting the foot arch, the heel contact point was relocated from the outside to the inside, and an additional contact point was placed on the inside of the toe, as shown in Figure 2. In addition, the vertical offset of the contact points relative to the ground was adjusted from 30 mm to 10 mm to prevent prediction errors when the GRF is calculated even when the foot is not in contact with the ground.

Using the total external force and the contact point positions, the vertical external force is distributed according to the contact point-ground surface distance to obtain the vertical GRF and center of pressure (COP) (i.e., frontal and sagittal GRMs). Finally, the horizontal GRFs are computed with the assumption that the GRF vector intersects the virtual pivot point (VPP) on the human body. Previous studies have experimentally reported that the GRF vector intersects the VPP, which is located above the COM of the human body [21]. In the proposed method, the horizontal GRF was calculated such that the GRF vector intersects the VPP, defined as 37.5 mm upward from the COM of the whole body.

**Figure 2 sensors-25-03705-f002:**
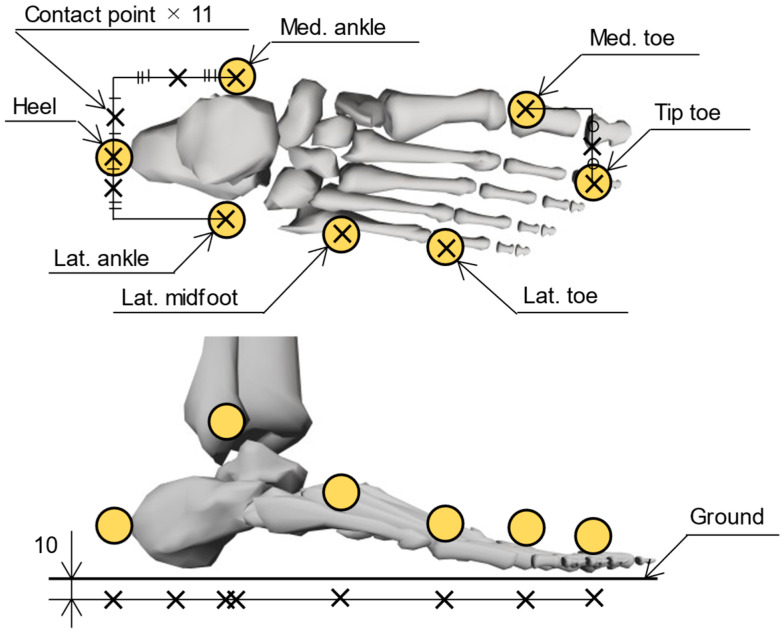
Placement of foot contact points. A total of 22 contact points, 11 per foot (indicated by black crosses), were located based on the reflective markers of the OMC system (yellow circles). The contact points were arranged with a 10 mm offset toward the ground during static standing.

### 2.2. Participants

The experiment was conducted with healthy participants, consistent with the conditions of previous studies, to verify the practicality of FDA estimation under footwear conditions by comparing its accuracy with that of the current biomechanical model-based method. Participants were 10 volunteers (7 males, 3 females; mean age: 23.0 ± 1.5 years; mean height: 1.66 ± 0.09 m; mean weight: 61.6 ± 11.9 kg) with no history of musculoskeletal disorders. This study was conducted in accordance with the Declaration of Helsinki and approved by the Ethics Committee of Tokyo Metropolitan University (protocol code H5-51). All participants received both verbal and written explanations regarding the study, and they provided written informed consent.

### 2.3. Conditions

This study conducted level-ground walking trials at three speeds (normal, fast, and slow), which were previously found to achieve practical estimation accuracy by the FDA under barefoot conditions. The length of the walking path was set to approximately 10 m. The experiment aimed to investigate differences in estimation accuracy between barefoot and footwear conditions. Sneakers and running shoes, commonly used in daily life and sports activities, were employed as the footwear. Participants wore their own running shoes and sneakers that they were accustomed to wearing to allow for variations within the categories of sneakers and running shoes. The running shoes had a synthetic fiber upper and a rubber sole (mean sole thickness: 44 ± 10 mm), while the sneakers had an upper made of artificial or natural leather and a rubber sole (mean sole thickness: 38 ± 9 mm). Normal speed was defined as each participant’s preferred walking pace, while the fast and slow speeds were approximately 120% and 80% of the normal walking speed, respectively. Each participant performed three trials per condition and familiarized themselves with each condition before the experiment.

### 2.4. Measurements

An OMC system (OptiTrack Flex3; Natural Point Inc., Corvallis, OR, USA) recorded the three-dimensional coordinates of markers attached to the whole body. The marker set included a total of 49 reflective markers based on the Gait 2392 musculoskeletal model in OpenSim [17,18,19,20]. The marker data were digitally filtered (low-pass filter, Butterworth fourth-order type, −3 dB at 6 Hz) and sampled at 100 Hz.

To validate the GRF prediction, a force plate (TF-4060-D; Tec Gihan Co., Ltd., Kyoto, Japan) measured the GRF and GRM. Participants walked multiple times under each experimental condition, and three trials in which they firmly contacted the force plate were used for data analysis. The baseline of the force plates was reset before each trial. The GRF and GRM data were digitally filtered (low-pass filter, Butterworth fourth-order type, −3 dB at 18 Hz) and sampled at 1000 Hz.

### 2.5. Data Analysis

The predicted GRFs and GRMs were compared with the measurement data from the force plate. The agreement between the predicted and measured waveforms was assessed using Pearson’s correlation coefficient (*ρ*), categorized as follows: *ρ* ≤ 0.35 for weak, 0.35 < *ρ* ≤ 0.67 for moderate, 0.67 < *ρ* ≤ 0.9 for strong, and 0.9 < *ρ* for excellent correlation [22]. In addition, prediction errors were quantitatively evaluated using the root-mean-square error (RMSE) and relative RMSE (rRMSE), which normalizes the RMSE by the average peak-to-peak amplitude for both solutions [23]. These indicators were computed by MATLAB 9.14 (MathWorks, Inc., Natick, MA, USA).

The average RMSE and rRMSE of three trials for each condition was used for statistical analysis. The nparLD package [24] in R, which performs nonparametric two-way repeated measures analysis of variance (RM-ANOVA), was selected for the statistical analysis because the residuals were confirmed to be non-normally distributed by the Shapiro–Wilk test. Significant main effects were further explored using Dunn’s post hoc test with the Holm–Bonferroni adjustment.

To investigate the discriminator ability of the proposed method, the peak vertical and anterior GRFs—commonly used in biomechanical analyses [2,25]—were compared across the three walking speed conditions. A one-way RM-ANOVA with Greenhouse–Geisser correction was employed, as the residuals were confirmed to be normally distributed by the Shapiro–Wilk test, but the assumption of sphericity was not confirmed by the Mauchly’s test. Significant main effects were further explored using multiple comparison with Holm–Bonferroni correction. These analyses were performed separately for both the estimated and measured GRFs.

This study also investigated the influence of participant characteristics on the RMSE and rRMSE. For items confirmed to follow a normal distribution by the Shapiro–Wilk test, a paired t-test was performed for gender analysis, and Pearson’s correlation coefficient was used for the analysis of height and weight. For the other items, the Wilcoxon signed-rank test and Spearman’s rank correlation coefficient were used for the analyses. All statistical analyses were conducted using R version 4.4.1, with *p* < 0.05 indicating statistical significance.

## 3. Results

The GRF and GRM curves throughout one gait cycle under normal speed conditions are presented in Figure 3. Across all experimental conditions, the predicted results showed excellent or strong correlations with the force plate measurements in the antero-posterior GRF (*ρ* = 0.81–0.90), vertical GRF (*ρ* = 0.87–0.97), sagittal GRM (*ρ* = 0.84–0.94), and transverse GRM (*ρ* = 0.69–0.83). Although the estimation accuracy for some GRFs/GRMs was lower than that reported in previous studies due to the simpler computation algorism employed in the proposed method, this study demonstrated an estimation accuracy comparable to that of previous biomechanical model-based approaches [9,11], as shown in Figure 4. Furthermore, the FDA successfully estimated GRFs and GRMs without the need for collecting training data and with reduced computational cost, achieving a computational time of approximately 4 s per trial (CPU: Intel Core i7-10700 with 4.0-GHz average speed; memory: 32 GB; software: MATLAB 9.14 and OpenSim 4.1; OS: Windows 11 Home). Thus, this study confirmed the practical accuracy and computational efficiency of the FDA for GRF estimation under footwear conditions.

Relatively lower correlation coefficients, including moderate correlations, were observed for the medio-lateral GRF (*ρ* = 0.64–0.79) and frontal GRM (*ρ* = 0.56–0.81). Statistical analysis revealed that the rRMSEs for medio-lateral GRFs and frontal GRMs were significantly affected by footwear conditions, with lower accuracy when shoes were worn (Table 1). For these items, the absolute error between the estimated and measured values was larger under footwear conditions, particularly during the loading response (early stance phase) and terminal stance (late stance phase), as shown in Figure 5. During these periods, foot arch deformation is strongly influenced by footwear characteristics, such as variations in the shape and material of the shoe sole, due to the large reaction forces acting on the foot. This negative effect significantly impacts the distribution of reaction forces generated at the foot contact points, especially in the medio-lateral COP, which varies with a narrow range, resulting in lower accuracy in the frontal GRM and medio-lateral GRF.

The RMSEs for all items were significantly influenced by walking speed, and the rRMSEs for antero-posterior and vertical GRFs and transverse GRMs were significantly affected by walking speed, with a decrease in accuracy observed at the high walking speed (Table 1). For these items, the absolute error between the estimated and measured values increased with walking speed during the stance phase, with a particularly large error during the loading response (early stance phase) and terminal stance (late stance phase), as shown in Figure 6. During these periods, the vertical GRF rises sharply at higher walking-speeds, leading to significant errors in the prediction based on foot deformation. Furthermore, while the FDA computes the horizontal GRF based on the assumption that the GRF vector from the COP always intersects the VPP, this assumption may not hold during the acceleration and deceleration phases of walking and lead to errors in the antero-posterior GRF. As a result, this mismatch can lead to errors in the antero-posterior GRF during high-speed walking.

In both the estimated and measured data, the peak vertical and anterior GRFs were significantly influenced by walking speed, with an increase in peak values at higher walking speeds (Figure 7). These results demonstrate that the proposed method has the discriminator ability to capture the effects of walking speed, consistent with the observations from force plate measurements.

The RMSEs for the anterior and vertical GRFs were significantly influenced by participant characteristics, with no significant differences for the other GRFs and GRMs. The RMSE for the anterior GRF in female participants was significantly higher than that in male participants (*p* = 0.007), and it increased significantly for shorter participants (*p* < 0.001, *ρ* = −0.88), and participants with lower weight (*p* = 0.008, *ρ* = −0.78), as shown in Figure 8. The RMSE for the vertical GRF increased significantly for shorter participants (*p* = 0.040, *ρ* = −0.66). The FDA might exhibit lower prediction accuracy when the foot deformation is small, as it estimates GRFs and GRMs based on the foot deformation caused by the load on the foot. This resulted in lower estimation accuracy for smaller participants.

**Figure 3 sensors-25-03705-f003:**
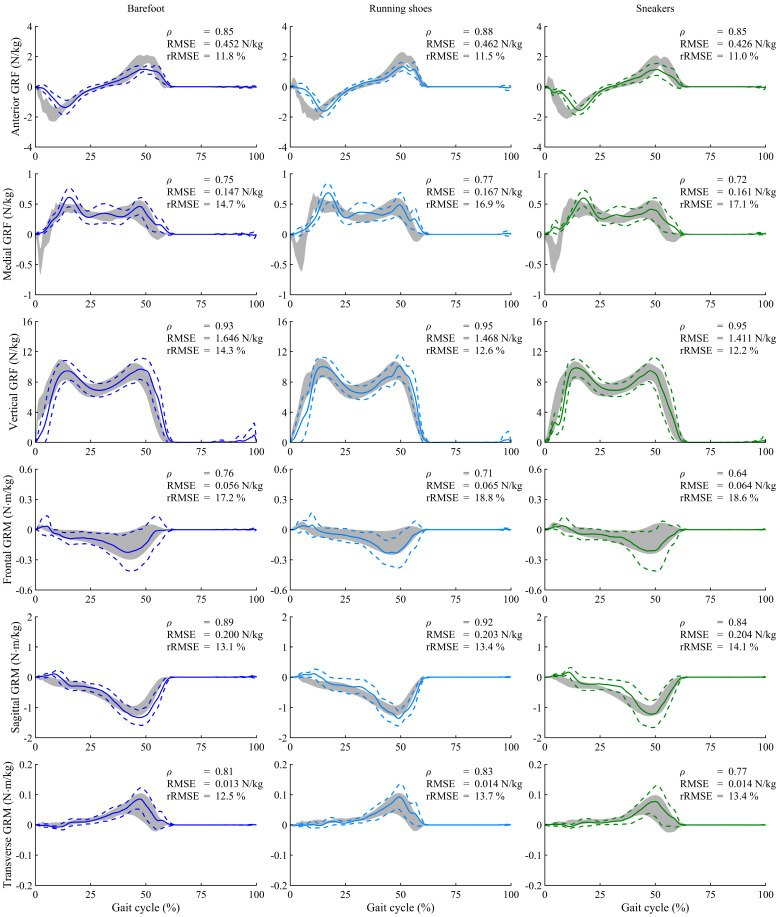
Ground reaction forces (GRFs) and moments (GRMs), normalized to the body mass of participants, during normal walking speed under three footwear conditions. Solid and dashed lines represent the predicted means and standard deviations for all participants, while gray shading indicates the measured values by the force plate. Each graph includes the Pearson’s correlation coefficient (*ρ*), root-mean-square error (RMSE), and the relative RMSE (rRMSE). The positive directions of GRFs are defined along the anterior, medial, and upward axes, and GRMs are represented as moments around these axes.

**Figure 4 sensors-25-03705-f004:**
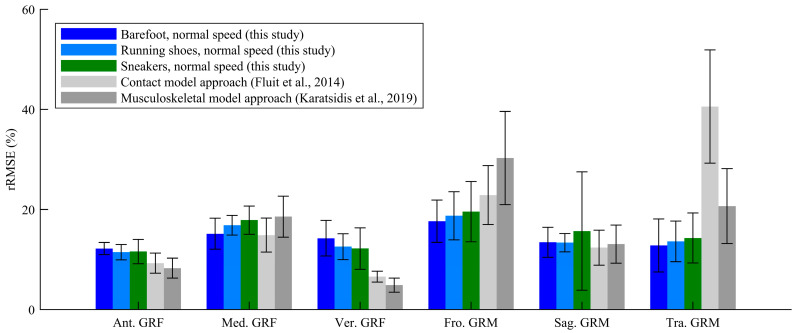
Relative root-mean-square errors (rRMSEs) of ground reaction forces (GRFs) and moments (GRMs) from this study and previous studies [9,11].

**Table 1 sensors-25-03705-t001:** Statistical analysis of two-way repeated measures analysis of variance in the root-mean-square error (RMSE) and the relative RMSE (rRMSE).

	Footwear Effect		Walking Speed Effect		Interaction	
	RMSE	rRMSE	RMSE	rRMSE	RMSE	rRMSE
Ant. GRF	*F* = 0.0742, *p* = 0.92	*F* = 0.310, *p* = 0.66	***F* = 78.8, *p* < 0.001 *** **Slow < Normal < Fast**	***F* = 14.5, *p* < 0.001 *** **Slow < Fast,** **Normal < Fast**	*F* = 0.468, *p* = 0.68	*F* = 1.79, *p* = 0.15
Med. GRF	*F* = 2.73, *p* = 0.072	***F* = 15.5,** ***p* < 0.001 *** **Bare < Running,** **Bare < Sneaker**	***F* = 62.7, *p* < 0.001 *** **Slow < Fast,** **Normal < Fast**	*F* = 1.96, *p* = 0.15	*F* = 1.18, *p* = 0.31	*F* = 1.84, *p* = 0.14
Ver. GRF	*F* = 1.79, *p* = 0.17	*F* = 1.83, *p* = 0.16	***F* = 24.6, *p* < 0.001 *** **Slow < Normal < Fast**	***F* = 11.8, *p* < 0.001 *** **Slow < Normal < Fast**	*F* = 0.765, *p* = 0.49	*F* = 0.416, *p* = 0.70
Fro. GRM	*F* = 1.89, *p* = 0.15	***F* = 3.71,** ***p* = 0.029 *** **Bare < Running**	***F* = 12.5, *p* < 0.001 *** **Slow < Fast,** **Normal < Fast**	*F* = 0.119, *p* = 0.88	*F* = 1.58, *p* = 0.19	*F* = 1.15, *p* = 0.32
Sag. GRM	*F* = 0.331, *p* = 0.70	*F* = 0.544, *p* = 0.58	***F* = 8.92, *p* < 0.001 *** **Slow < Fast,** **Slow < Normal**	*F* = 1.84, *p* = 0.17	*F* = 1.36, *p* = 0.26	*F* = 2.10, *p* = 0.11
Tra. GRM	*F* = 0.192, *p* = 0.77	*F* = 0.613, *p* = 0.54	***F* = 20.4, *p* < 0.001 *** **Slow < Fast,** **Normal < Fast**	***F* = 9.94, *p* < 0.001 *** **Normal < Fast**	*F* = 1.68, *p* = 0.18	*F* = 2.15, *p* = 0.093

Note: * indicates a significant main effect (*p* < 0.05). Significant differences for each condition in the post hoc test are indicated by <. No significant interaction effects were observed for any items.

**Figure 5 sensors-25-03705-f005:**
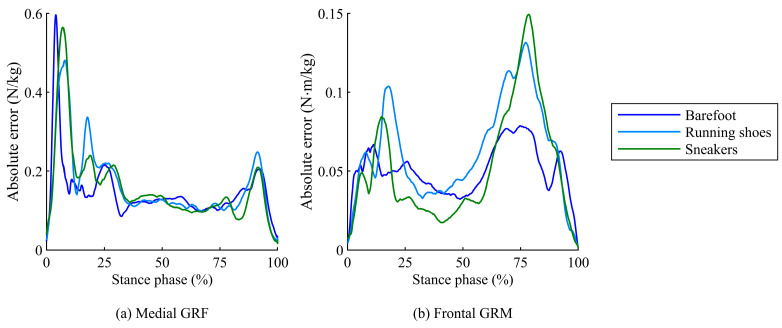
Time-series plots of absolute errors for (**a**) medio-lateral GRF and (**b**) frontal GRM between predicted and measured values during the stance phase under three footwear conditions.

**Figure 6 sensors-25-03705-f006:**
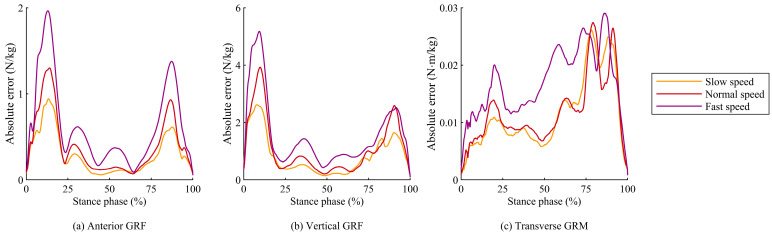
Time-series plots of absolute errors for (**a**) antero-posterior GRF, (**b**) vertical GRF, and (**c**) transverse GRM between predicted and measured values during the stance phase under three walking speed conditions.

**Figure 7 sensors-25-03705-f007:**
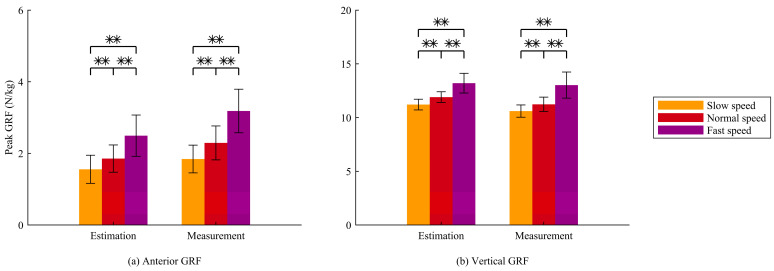
(**a**) Peak anterior GRF and (**b**) peak vertical GRF from estimated and measured data across three waking speeds. Significant differences between the conditions are indicated by ** (*p* < 0.01).

**Figure 8 sensors-25-03705-f008:**
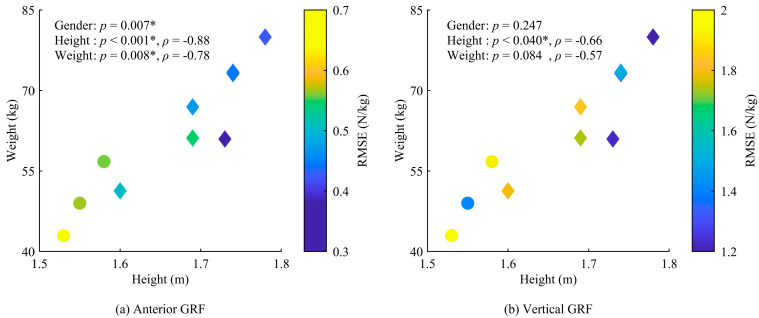
Relationship between estimation accuracy and participant characteristics, including gender, height, and weight, for (**a**) antero-posterior GRF and (**b**) vertical GRF. The circles indicate data for female participants, and the diamonds represent data for male participants. Each graph includes the correlation coefficient (*ρ*) and *p*-value. Significant effects are indicated by * (*p* < 0.05).

## 4. Discussion

This study investigated the prediction accuracy of the FDA, which estimates GRFs and GRMs using OMC recordings for biomechanical analysis. Previous studies have performed biomechanical analysis using OMC not only in a laboratory environment but also in clinical settings such as sports fields [26]. In contrast, biomechanical analysis with GRF measurements in clinical settings using force plates, which need to be installed on the ground, remains challenging due to the low portability, setup complexity, and financial costs associated with force plates. Our estimation method using the FDA enables GRF analysis using only OMC, thus enabling biomechanical analysis based on GRF measurements in clinical settings.

One advantage of the FDA is that it estimates GRFs and GRMs without relying on training data. While many previous studies have developed a machine learning-based approach for GRF estimation [5,6,7], it poses the challenge of high costs in collecting large amounts of training data. In contrast, the FDA estimates GRFs and GRMs based on biomechanical assumptions and foot deformations during human activities, an approach that does not require collecting training data. Therefore, the FDA has potential applicability for analyzing various activities, including special sports movements, and various participants, such as individuals with disabilities and the elderly.

Compared to current biomechanical model-based methods that eliminate dependence on training data, the FDA offers the advantage of estimating the GRF with fast computation. Previous studies involving the biomechanical model-based approach employed musculoskeletal and contact models including an optimization process [8,9,10,11,12,13], which led to high computational costs for computing GRFs. In contrast, the FDA eliminates the optimization process in GRF estimation by focusing on foot deformations during human activities. Therefore, the FDA has the advantage of rapid analysis and feedback for participants, contributing to the advancement of biomechanical model-based estimation in clinical settings.

The main finding of this study is that the FDA, which eliminates the need for training data and reduces the computational cost, enables GRF prediction under footwear conditions. Biomechanical analyses of human activities have been performed not only for barefoot conditions but also for footwear conditions [15,16]. Although a previous study confirmed that the FDA enables GRF estimation with bare feet [14], its estimation accuracy with shoed feet, which affect the characteristics of foot deformation, has not been investigated. In this study, the FDA under footwear conditions obtained practical prediction accuracy comparable to previous studies [9,11]. This finding suggests that the FDA is an effective method for estimating GRFs and GRMs in clinical situations where participants wear shoes.

In addition to the accuracy analysis, this study confirmed that the proposed method has the discriminator ability to capture practical evaluation factors in biomechanical analyses. Previous studies have employed peak vertical and anterior GRFs to assess human activities, such as evaluating physical ability in the elderly [25] and assessing injury risk in athletes [27]. In such situations, the proposed system—based solely on OMC—is effective for conducting the GRF analysis in clinical settings. Therefore, the proposed GRF estimation method contributes to providing useful information for biomechanical analysis in clinical settings.

Although the FDA enables GRF estimation under footwear conditions, the characteristics of footwear negatively affect the prediction accuracy. The FDA estimates GRFs and GRMs from multiple markers placed on foot segments based on the marker set of the Gait 2392 model in OpenSim, which has been found to be robust against marker misalignment on the order of a few millimeters [14]. However, the characteristic of footwear impacts the accuracy of the FDA, which estimates GRFs and GRMs based on the deformation of the skin and foot arch under load. In addition, the estimation accuracy decreased at high walking speeds because GRF and GRM values tend to rise as walking speed increases. These findings suggest that further investigation is needed for the use of specialized sports shoes or analyzing more aggressive movements.

For application to other shoes and movements, it may be necessary to reconsider the arrangement and number of contact points in the FDA. This study estimated GRFs and GRMs using the FDA with markers attached to the footwear, based on the assumption that the foot and shoes exhibit the same motion. Our FDA method minimizes the negative effects of errors between the foot and footwear motions by defining the placement and number of contact points using multiple markers. However, the prediction accuracy of GRMs is negatively impacted by variations in the moment arm due to changes in the position of the contact points. While this study adjusted the contact point locations to prevent the decrease in GRM estimation accuracy caused by the arch support of the shoe sole, it may be necessary to reconsider the contact point arrangement to maintain estimation accuracy for other types of footwear with uneven sole shapes. In addition, increasing the number of contact points may be necessary for accurate GRM estimation in the analysis of other movements, such as jumping, where the COP is located at the edge of the foot.

This study has several limitations. First, the FDA might exhibit lower prediction accuracy for participants with smaller body sizes because their foot deformation is smaller. In this study, significant correlations between participant characteristics and prediction accuracy were found for only a few GRFs due to the small sample size. However, further investigation is needed to assess the applicability of the FDA as a practical estimation method for participants with smaller bodies, such as children. Second, the FDA cannot achieve good estimation accuracy on uneven ground, because it estimates GRFs and GRMs based on the distance between the markers attached to the foot and the ground surface. Lastly, the statistical analysis in this study may have resulted in a type II error for items where no significant differences were found, as the small sample size led to insufficient statistical power for these items. Therefore, these less robust results should be interpreted carefully.

## 5. Conclusions

Our approach for estimating GRFs and GRMs based on foot deformations successfully achieved practical accuracy for level walking with running shoes or sneakers. This prediction method, with its low computational cost and lack of requirement for training data, offers advantages for analyzing various activities with footwear, contributing to the versatility of GRF prediction. However, it should be noted that the applicability of this method to more aggressive sports activities with specialized footwear needs further investigation, as the FDA cannot fully eliminate the impact of shoes and movement speed on prediction quality.

## Figures and Tables

**Figure 1 sensors-25-03705-f001:**
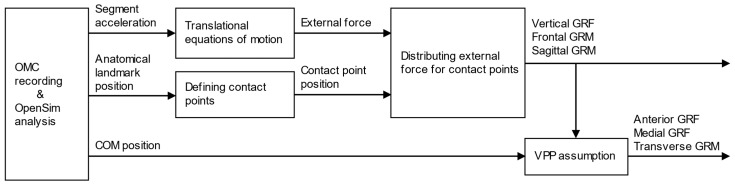
Computational flow for estimation of ground reaction forces (GRFs) and moments (GRMs). The vertical GRF and frontal and sagittal GRMs are calculated by distributing the external force to each contact point defined on the foot segment. The horizontal GRFs and transverse GRM are then calculated based on the assumption that the GRF vector intersects the virtual pivot point (VPP) on the human body.

## Data Availability

The data presented in this study are available as Appendix A.

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
