# Peer review of "Verification of Footwear Effects on a Foot Deformation Approach for Estimating Ground Reaction Forces and Moments"

_sensors, 2025, doi:10.3390/s25123705_

Round 1

Reviewer 1 Report

Comments and Suggestions for Authors

General Comments

In the current manuscript, the authors analyze the agreement of the foot deformation approach (FPA) with estimates of ground reaction forces and moments during ground walking while the subjects are walking with shoes, rather than barefoot as in their previous study. The overall manuscript is well written, and the proposed analyses are correctly performed.

Nonetheless, the manuscript can be improved in some parts by performing some further analyses.

Methodologically, RMSE and RMSEr alone seem not sufficient for assessing agreement in this specific case. What about assessing also the correlations between reference and estimates on the time-normalized GRFs and GRMs? It will give even more insights about the quality of your methodology and to its specific application in this context.

Once you reported these results, it would be of great help if you are able to also define if your model is able to capture minimal meaningful information in clinical context. For instance, is your GRF estimate able to capture some important features of a gait cycle that can be embedded in further clinical analyses? Or: do your estimates consistently differ in some portion of the gait cycle that is more meaningful than others?

Specific Comments

Introduction

Line 32—25: What do you mean by this? I feel you are referring to pressure insoles when you say wearables, but it is not made explicit. Please, consider specifying to which wearable system you are referring to.

Line 46—47: This is a bold statement, and I personally feel you should justify it with appropriate references.

Line 50—52: To which method are you referring to? It would be beneficial for the reader to have a reference at this point of the text.

Materials and Methods

Line 83: I would rephrase as ‘In this study we estimate’ or ‘In this study GRFs … have been estimated’.

Line 123: I think you should report the approval number of the Ethics Committee, along with this statement.

Line 127: What was the distance they walked? This seems to me an important information to report. Please, expand on this.

Line 145—147: Did you do something for resetting the baseline of the force plates from trial-to-trial or from participant-to-participant? Please, expand on this. Moreover, how did the participants hit the force plate? Did you consider a good trial only the ones where the foot was completely on the force plate only? Please, clarify the information about this methodological aspect.

Results

Line 167—168: On the basis of what you state that the agreement was good? Please, expand on this by citing numerical results.

Line 172—177: The two sentences seem redundant. Please, consider to reduce to one statement only.

Line 178—179: You must report some number at this stage of the text (i.e., the results section). For each of your statement, there should be a corresponding metrics that justifies what you say. I strongly recommend to follow this criterion and be consistent with it throughout all the results section.

Figure 3: I feel the y-labels are not reported correctly in this figure. You should report the quantity on the y-axis, not the plane, which in turn may be reported elsewhere (e.g., in the white spaces that you have on each subplot). Please, correct the y-label accordingly.

Table 1: This table, although presenting exhaustive information, is difficult to read. I feel it would be helpful for the reader if you present these results using some graphical method, e.g., a bar plot.

Discussion

Line 261—262: Are we sure that, although the accuracy may be good, the metrics that you want to see are actually clinically meaningful? This consideration of mine comes from the general comments above.

Reviewer 2 Report

Comments and Suggestions for Authors

Generally, a well-written, compact, report on examining effects of footwear to the FDA methods previously developed by the authors.

In the abstract (on line 17) it is unclear at first glance which item concerns your previous study, and which items the present study. Please clarify the abstract.

The concept VPP is not introduced to the reader. I needed to check it in the previous study [14]. Please and the few words of explanation so that a reader with no immediate access to prev. paper can follow.

The terminology of "running shoes" and "sneakers" is not that aligned globally, thus it would be a good addition to describe specifications of the shoes, eg. sole thicknesses. The current description in lines 131-133 is fine, if you didn't collect detailed data on shoes. Maybe, OMC + Opensim modelling can give you estimate on how high from the floor calcaneuos was in each trial? A sample photo of the shoes could help reader as well.

In Figure 1, why there is concept "Translational equations of motion", aren't the rotational motions included as well?

In my opinion, the manuscript need to be improved by  adding graphical and statistical results. Specifically, I have found statistical parameter mapping a clarifying statistical method for time series data in motion analysis (see. eg. https://spm1d.org/ ).  I see that in this study, SPM will reveal the statistical significance of the difference between OMC measurements and the estimates on the stance phase timeline. This should be doable for the authors, as they have advanced statistical analyses already in the manuscript. If authors have made a deliberate decision of not using the 1-D SPM, also happy to hear a justification.

Further, The manuscript body now show only the normal speed, running shoes results in the graph, and supplement shows data in table. At least, the supplementary material should include graphs of all three walking speeds and both footwear, (preferably with spm results).

The manuscript body could include normal speed results for both footwear. The current mode of only showing one of the footwear raises a questions if only "the better results" are being shown. 

In Figure 3 caption please introduce the meaning of the notations in the graphs firstly (or make a "legend" box).

Discussion is very good, and includes limitations of the study acknowledged.

Round 2

Reviewer 2 Report

Comments and Suggestions for Authors

The authors have addressed my comments well, with good justifications on their selections in statistical analyses.